# A Versatile Quantum Simulator for Coupled Oscillators Using a 1D Chain of Atoms Trapped near an Optical Nanofiber

**Daniela Holzmann *, Matthias Sonnleitner**  **and Helmut Ritsch**

Institute for Theoretical Physics, University of Innsbruck, Technikerstraße 25, A-6020 Innsbruck, Austria;
matthias.sonnleitner@uibk.ac.at (M.S.); helmut.ritsch@uibk.ac.at (H.R.)
* Correspondence: daniela.holzmann@uibk.ac.at

**Abstract:** The transversely confined propagating light modes of a nanophotonic optical waveguide or nanofiber can effectively mediate infinite-range forces. We show that for a linear chain of particles trapped within the waveguide's evanescent field, transverse illumination with a suitable set of laser frequencies should allow the implementation of a coupled-oscillator quantum simulator with time-dependent and widely controllable all-to-all interactions. Using the example of the energy spectrum of oscillators with simulated Coulomb interactions, we show that different effective coupling geometries can be emulated with high precision by proper choice of laser illumination conditions. Similarly, basic quantum gates can be selectively implemented between arbitrarily chosen pairs of oscillators in the energy as well as in the coherent-state basis. Key properties of the system dynamics and states can be monitored continuously by analysis of the out-coupled fiber fields.

**Keywords:** fiber optics; quantum simulation; collective light scattering

---

## 1. Introduction

Suitably designed laser fields allow one to trap individual quantum particles at well defined locations and cool them to their motional ground states [1,2]. Some time ago, it was demonstrated that trapping and cooling is also possible close to optical nanostructures and, in particular, in the vicinity of a tapered optical nanofiber [3–5]. More recently, Meng at al. successfully scattered light from a transverse excitation laser into a nanofiber using atoms trapped near that fiber [6].

Once trapped, the atoms interact with the evanescent field of light modes propagating within the fiber [5], exchanging energy and momentum. Thus, the light strongly influences the atomic motion in the trap which in turn modifies the light propagation [7–10]. As photons within the fiber propagate over practically infinite distances, they collectively couple to all atoms, which induces all-to-all long-range interactions [11]. In this way thousands of atoms can be trapped, which leads to strong collective effects [12].

The individual atom–atom coupling via resonant photon emission by one atom followed by absorption by a second atom is typically rather small [13,14], but it can already lead to spatial self-ordering of the atoms [15]. The induced force can be significantly increased if the atoms are transversely illuminated far off any internal atomic resonance to induce collective coherent scattering into the fiber [16–18]. Here, the interference between the mode amplitudes created by scattering from different particles leads to gradient or dipole forces, which appear without changing the internal atomic state from spontaneous emissions [19]. The properties of these forces can be modified by the help of two laser frequencies [20].

The interactions between the particles depend on the properties of the incoming light field. With careful choice of laser frequencies and powers, almost arbitrary shapes of interaction forces can be synthesized [21]. In this work, we give examples of how this property could be used for quantum simulation [22–30], as well as for quantum computation. By designing the incoming light field, we show, for example, how the

interaction between ions can be simulated, even if they are ordered in 2D or 3D geometries. In contrast to quantum simulation with ions [31,32], we can even turn off the interactions between arbitrary pairs of particles. In the second part, we describe the oscillator states as qubits and use this approach to design quantum gates [33,34] or produce entangled states [35–37].

## 2. Materials and Methods

### 2.1. Tailored Coupling of the Quantized Motion of a Trapped Atom Chain

In this work, we consider $N$ particles, typically atoms, molecules or nanospheres, harmonically trapped at predefined positions along an optical nanofiber. As depicted in Figure 1, these atoms interact with the evanescent field of the propagating nanofiber modes. Additionally, the particles are transversely illuminated by pump fields of tunable frequencies. Each particle thus coherently scatters light from the pump fields into the fiber, where it interferes with light scattered by other atoms. The particles thus redistribute the field along the fiber, which leads to effective interactions and forces between the particles. The interaction created by each frequency component of the pump light is long-range and depends on the distances between pairs of particles on the wavelength scale. Hence, displacing one particle changes the overall fiber field and thus the forces acting on all other particles.

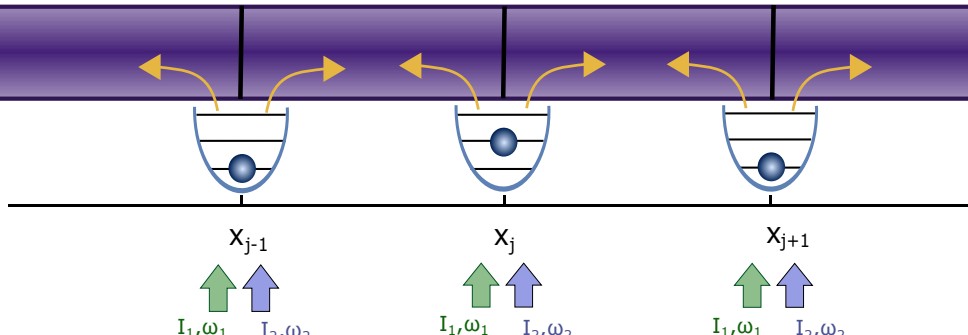

**Figure 1.** Sketch of our system: $N$ particles are confined in homogeneous traps next to a nanofiber. The particles are illuminated by multi-color transverse pump fields and scatter light into the fiber. Interference of the scattered fields in the fiber leads to effective forces between the particles.

Similar long-range interactions and forces have been discussed already in a pioneering work by Chang et al. in [16]. There the focus was on resonant excitation and radiation pressure induced by internal transitions of an atom coupled to the waveguide. In line with our previous work [21], we will allow for a very general form of mechanical interaction between the particles which can be achieved via frequency shaping of the illumination light.

In our model, the transverse pump field is a sum of many plane waves with different intensities and frequencies. We assumed that the different spectral components are sufficiently distinct such that the interference terms are negligible as the individual components are spatially coherent but not time-coherent and interactions average out over time. The effective pair forces between the particles in such a system can be calculated using a beamsplitter matrix model describing all the scattering processes by the particles [19].

Note that the particles, in principle, also back scatter a fraction of the field propagating inside the fiber. However, if we assume weak coupling of the particles to the fiber [16,21], each particle reflects only a tiny fraction of the propagating fiber fields. Of course, this assumption also requires a small scattering rate from the transverse beam into the fiber, but this can be balanced by using large powers of the incoming field. This is why we neglect back scattering effects and assume that the force on the particles arises solely due to interference effects of the fields scattered into the fiber from the transverse pump. This assumption generally works well for small particle numbers, but for large system sizes even a small reflection by each particle can lead to significant collective effects [2].

Within this approximation, the radiation force $F_j$ on a classical particle at position $x_j$ can be written as a sum of effective two-particle forces $f_{\text{pair}}(x_i, x_j)$ between this particle and all the other particles at positions $x_i$ [21]

$$F_j = \sum_{\substack{i=1 \\ i \neq j}}^{N} f_{\text{pair}}(x_i, x_j) = \sum_{\substack{i=1 \\ i \neq j}}^{N} \sum_{k} \frac{\sigma_{\text{sc}} I_k \cos\left(k(x_j - x_i)\right) \text{sign}(x_i - x_j)}{c}, \tag{1}$$

with $N$ the number of particles along the fiber, $I_k$ the intensity of the field with frequency $\omega_k = k/c$ and $\sigma_{\text{sc}}$ the scattering cross section between the particles and the beam.

Using this force, we defined a two-particle potential $u_{\text{pair}}(x_i, x_j)$ such that $f_{\text{pair}}(x_i, x_j) = -\partial_{x_j} u_{\text{pair}}(x_i, x_j)$. For a system of $N$ particles, the total potential is thus the sum of all two-particle interactions

$$U_{\text{tot}}(x_1, \ldots, x_N) = \frac{1}{2} \sum_{j=1}^{N} \sum_{\substack{i=1 \\ i \neq j}}^{N} u_{\text{pair}}(x_i, x_j) = \frac{1}{2} \sum_{j=1}^{N} \sum_{\substack{i=1 \\ i \neq j}}^{N} \sum_{k} \frac{\sigma_{\text{sc}} I_k}{ck} \sin\left(k|x_i - x_j|\right). \tag{2}$$

It is clear that any translation of one particle changes the light field along the fiber and thus the optical potential seen by all the other particles. The same result can be obtained following the model of Chang et al. [16] by taking the weak scattering limit for far-detuned light. Eliminating the internal excited states of the particles then leads to the force given in Equation (1).

We studied such a system in a previous work [21] where we assumed classical point particles allowed free movement along the fiber direction. Here, we study locally trapped and very cold particles, which requires a quantized description of motional degrees of freedom.

In the present model, we thus considered particles trapped in harmonic potentials centered at positions $x_{j,0}$, $j = 1, \ldots, N$, $U_{\text{HO}} = \sum_{i=1}^{N} m\omega_T \Delta_i^2/2$, with $m$ being the mass of the particles, $\omega_T$ being the frequency of the harmonic oscillator traps and $\Delta_i = x_i - x_{i,0}$. We assumed that the particles were tightly trapped in the transverse direction and linearized the motion along the longitudinal motion of the particles in Equation (2) around the center of the harmonic oscillators $x_{i,0}$, $x_i \to x_{i,0} + \Delta_i$, with $k\Delta_i \ll 1$. We thus obtained an effective Hamiltonian

$$\hat{H} = \sum_{k} \frac{I_k \sigma_{\text{sc}}}{c} \sum_{j=1}^{N} \left( \sum_{i=1}^{j-1} \left( \frac{1}{k} \sin\left(kd_{ij}\right) + (\Delta_j - \Delta_i) \cos\left(kd_{ij}\right) \right) \right.$$
$$\left. - \sum_{i=1}^{N} \frac{k(\Delta_j - \Delta_i)^2}{4} \sin\left(kd_{ij}\right) \right) + \sum_{i=1}^{N} \left( \frac{P_i^2}{2m} + \frac{m\omega_T^2}{2} \Delta_i^2 \right), \tag{3}$$

with $d_{ij} = |x_{j,0} - x_{i,0}|$. We quantize the relative motion of the particles with respect to the trap centers by setting

$$\hat{\Delta}_i = \sqrt{\frac{\hbar}{2m\omega_T}} (\hat{a}_i + \hat{a}_i^\dagger) = \delta_0 (\hat{a}_i + \hat{a}_i^\dagger), \tag{4a}$$

$$\hat{P}_i = i\sqrt{\frac{\hbar m \omega_T}{2}} (\hat{a}_i^\dagger - \hat{a}_i) = \frac{i\hbar}{2\delta_0} (\hat{a}_i^\dagger - \hat{a}_i), \tag{4b}$$

with the oscillator length $\delta_0^2 = \hbar/(2m\omega_T)$.

Ignoring the constant terms in the Hamiltonian we thus obtain

$$\hat{H} = \hat{H}_{\text{osc}} + \hat{H}_{\text{int}} + \hat{H}_{\text{rwa}} \tag{5a}$$

$$\hat{H}_{\text{osc}} = \sum_{i=1}^{N} \hbar \left( \omega_T - \sum_{j=1}^{N} \sum_{k} \Omega_k \sin(kd_{ij}) \right) \hat{a}_i^\dagger \hat{a}_i = \sum_{i=1}^{N} \hbar \tilde{\omega}_i \hat{a}_i^\dagger \hat{a}_i \tag{5b}$$

$$\hat{H}_{\text{int}} = \sum_{j=1}^{N} \sum_{i=1}^{N} \sum_{k} \hbar \Omega_k \sin(kd_{ij}) \hat{a}_j^\dagger \hat{a}_i \tag{5c}$$

$$\hat{H}_{\text{rwa}} = \hbar \sum_{j=1}^{N} \left( \sum_{i=1}^{j-1} \sum_{k} \epsilon_k \cos(kd_{ij}) \left( \hat{a}_j + \hat{a}_j^\dagger - \hat{a}_i - \hat{a}_i^\dagger \right) \right.$$

$$\left. - \sum_{i=1}^{N} \sum_{k} \frac{\Omega_k}{2} \sin(kd_{ij}) \left( \hat{a}_j^2 + \hat{a}_j^{\dagger 2} - \hat{a}_j \hat{a}_i - \hat{a}_j^\dagger \hat{a}_i^\dagger \right) \right), \tag{5d}$$

with $\Omega_k := \sigma_{\text{sc}} I_k \delta_0^2 k / (\hbar c)$ and $\epsilon_k := \sigma_{\text{sc}} I_k \delta_0 / (\hbar c)$. Here, $\hat{H}_{\text{rwa}}$ generates force terms in the time evolution oscillating at the trapping frequency or higher, which typically cancel out when averaged over one period. Hence, we will neglect them later in Section 3.2. Effectively they lead to a small displacement of the particles equilibrium and a change of the effective frequency of the oscillators (squeezing). $\hat{H}_{\text{int}}$ describes the interactions between the particles and $\hat{H}_{\text{osc}}$ the harmonic potential with shifted frequency $\tilde{\omega}_i$ due to the interaction of the particles.

### 2.2. Model Assumptions and Limitations

In our model, we assumed that the particles are harmonically trapped and we require that the transverse pump fields convey strong enough particle–particle interactions to influence the motion of the trapped particles along the fiber direction. At the same time, the fields must be weak enough such that we can neglect saturation effects and eliminate any particle's internal degrees of freedom. In the following, we study in more detail how all these limiting conditions restrict the operating parameters.

The intensity of the light field scattered into the fiber by the particles depends on the incoming photon energy $\hbar\omega$ and flux, the emission rate into the fiber $\gamma_{\text{guid}}$, the effective mode cross section $A$ of the fiber field and the excited state population $\rho_{ee}$. We approximated the particles as effective two-level systems and operate at low saturation by choosing large laser detuning $\Delta \gg \Gamma$ from atomic resonance, with $\Gamma$ as the decay rate. In this limit, the excited state fraction is $\rho_{ee} \approx I_0 / (2I_{\text{sat}}(1 + 4\Delta^2/\Gamma^2)) \ll 1$, with $I_0$ the intensity of the incoming pump field and $I_{\text{sat}}$ the saturation intensity.

The emission rate into the fiber depends on the spatial profile of the fiber field determining the field strength at the atomic position and on the atomic dipole matrix element. A single-mode fiber carries only the fundamental $\text{HE}_{11}$-mode. The explicit expression for the mode profiles are given in [38]. Here, we assume that the modes and the atomic dipoles are linearly polarized, perpendicular to the fiber axis. In this case, the particles scatter the light symmetrically into the fiber. Using Fermi's golden rule, the emission rate into the fiber can be found with $\gamma_{\text{guid}} \approx 0.13\,\Gamma$ and with $\Gamma = |\bar{d}|^2 \omega^3 / (3\pi\epsilon_0 \hbar c^3)$, the free-space emission rate [39]. Usually $\gamma_{\text{guid}}$ for atoms along a nanofiber is between 0.1 and 0.2 $\Gamma$ [39], but can be tuned up to 0.99 $\Gamma$ for quantum dots [40,41] or superconductory transmon qubits [42]. Recently, the coupling efficiency has been improved for atoms when using a hole-tailored nanofiber and reached 0.6 $\Gamma$ [43].

The intensity scattered into the fiber by the particles can thus be estimated by

$$I_k = \frac{\hbar\omega}{A} \gamma_{\text{guid}} \rho_{ee} \approx \frac{1}{2} \frac{\hbar\omega}{A} \gamma_{\text{guid}} \frac{I_0/I_{\text{sat}}}{1 + 4\Delta^2/\Gamma^2}. \tag{6}$$

For a nanofiber with a radius $r = 200$ nm, the fiber cross section area $A = r^2\pi$, the Cesium D2-line $\omega \approx 2.2 \times 10^{15}$ Hz with $\Gamma = 33 \times 10^6$ 1/s and the mass of cesium $m \approx 220 \times 10^{-27}$ kg, and a detuning $\Delta = 100\,\Gamma$, we found that the intensity scattered into the fiber is $I_k \approx 6.2 \times 10^{-4}\, I_0/I_{\text{sat}}$ W/m$^2$.

To linearize the interaction potential requires a very deep potential such that the particles are well trapped $k\Delta_i \ll 1$. With $\Delta_i = \delta_0(a_i^\dagger + a_i)$ and $\delta_0 = \sqrt{\hbar/(2m\omega_T)}$, this means

$$k\delta_0 = \frac{\omega}{c}\sqrt{\frac{\hbar}{2m\omega_T}} \ll 1 \tag{7}$$

and we find

$$\omega_T \gg \frac{\omega^2}{c^2}\frac{\hbar}{2m}. \tag{8}$$

Using the parameters for the Cesium D2-line this implies $\omega_T \gg 10^5$ Hz.

Using these requirements, we can compare the trapping potential $H_T$ with the interaction arising from the transverse pump $H_{\text{pump}}$. Starting from the initial potential $\hat{H} = \hat{H}_{\text{pump}} + \hat{H}_T$ without any expansion

$$\hat{H} = \frac{1}{2}\sum_{j=1}^{N}\sum_{l=1}^{N}\frac{\sigma_{\text{sc}}I_k}{ck}\sin(kd_{jl}) + \sum_{j=1}^{N}\hbar\omega_T \hat{a}_j^\dagger \hat{a}_j, \tag{9}$$

we found $\hat{H}_{\text{pump}} \propto \sigma_{\text{sc}}I_k/(2kc)$ and $\hat{H}_T \propto \hbar\omega_T$. The scattering cross section can be approximated by the fiber cross section $\sigma_{\text{sc}} \approx A$. Inserting the intensity from Equation (6) and the boundary on the trapping frequency from Equation (8),

$$\frac{\hbar\omega_T}{\frac{\sigma_{\text{sc}}I_k}{2kc}} \approx \frac{4\omega_T}{\gamma_{\text{guid}}\rho_{ee}} = \frac{8\omega_T}{\gamma_{\text{guid}}}\frac{1 + 4\Delta^2/\Gamma^2}{I/I_{\text{sat}}} \gg \frac{8}{\gamma_{\text{guid}}}\left(\frac{\omega}{c}\right)^2\frac{\hbar}{2m}\frac{1 + 4\Delta^2/\Gamma^2}{I/I_{\text{sat}}}. \tag{10}$$

Using again the parameters for cesium and setting $\rho_{ee} \approx 10^{-2}$ and $\omega_T \approx 10^6$ Hz, we obtained the condition $\hbar\omega_T/(\sigma_{\text{sc}}I_k/(2kc)) \gg 3$.

Assuming tightly trapped particles, the interaction strength in the Hamiltonian (5a) is thus limited by

$$\Omega_k = \frac{\sigma_{\text{sc}}I_k\delta_0^2\omega}{\hbar c^2} \ll \frac{\gamma_{\text{guid}}I/I_{\text{sat}}}{2(1 + \Delta^2/\Gamma^2)}. \tag{11}$$

Using the parameters for Cesium given above, we found $\Omega_k \ll 10^5$ Hz.

## 3. Results

The Hamiltonian in Equation (5a) shows that we can design the effective interactions between the particles by choosing the intensities and frequencies of the incoming fields as well as the distances between the trapping positions of the particles. In the following section, we show how this can be used to simulate a system where particles interact via some specific physical potential of choice. Here, as a generic long-range interaction, we chose a Coulomb-type $1/r$ potential. In Sections 3.2 and 3.3, we discuss how this approach can be used to design quantum gates or how to entangle the motion of many particles.

### 3.1. Simulating Coulomb Interactions between Trapped Quantum Particles

The Hamiltonian in Equation (5a) can be used to simulate any symmetric two-body interaction. In the following example, we will use the effective atom–atom interaction via the waveguide to mimic the Coulomb interaction between ions.

In principle, one could tune the light fields to mimic a full Coulomb potential, but since $1/r$ is difficult to approximate in a Fourier series, this would require a very large number of laser fields. However, if we assume that the ions are also harmonically trapped, we only have to tune the atom–light interaction to mimic the Coulomb interaction at the position of the trapped ions.

To simulate the Coulomb (or any other) potential, we first linearized it around the trapping positions, quantized the motional degrees of freedom and then compared the terms with the Hamiltonian from Equation (5a). Using this concept, one can even map higher-dimensional systems of interacting particles to our 1D-system.

Figure 2 shows an example where we simulated the interaction between three ions distributed along a line by a system of particles along a nanofiber. The Coulomb potential $V_{\text{coul}}$ of $N$ interacting ions along a line of charge $q_i$ at distances $D_{ij}$ is given by

$$V_{\text{coul}} = \sum_{i=1}^{N} \sum_{j \neq i} \frac{1}{8\pi\epsilon_0} \frac{q_i q_j}{D_{ij}}. \tag{12}$$

Linearizing around their trapping position, quantizing the motional degrees of freedom and ignoring the constant terms, we find $q = q_i = q_j$.

$$\hat{H}_{\text{coul}} = \hat{H}_{\text{coul}_{\text{osc}}} + \hat{H}_{\text{coul}_{\text{int}}} + \hat{H}_{\text{coul}_{\text{rwa}}}, \tag{13a}$$

$$\hat{H}_{\text{coul}_{\text{osc}}} = \sum_{i=1}^{N} \hbar \left( \omega'_T + \frac{1}{8\pi\epsilon_0\hbar} \sum_{j \neq i} \frac{4q^2 \delta_0'^2}{D_{ij}^3} \right) \hat{a}_i^\dagger \hat{a}_i, \tag{13b}$$

$$\hat{H}_{\text{coul}_{\text{int}}} = -\frac{1}{8\pi\epsilon_0} \sum_{i=1}^{N} \sum_{j \neq i} \frac{4q^2 \delta_0'^2}{D_{ij}^3} \hat{a}_i^\dagger \hat{a}_j, \tag{13c}$$

$$\hat{H}_{\text{coul}_{\text{rwa}}} = \frac{1}{8\pi\epsilon_0} \sum_{j=1}^{N} \left( \sum_{i=1}^{j-1} \frac{-2q^2 \delta_0'}{D_{ij}^2} \left( \hat{a}_j + \hat{a}_j^\dagger - \hat{a}_i - \hat{a}_i^\dagger \right) \right.$$
$$\left. + \sum_{j \neq i} \frac{2\delta_0'^2 q^2}{D_{ij}^3} \left( \hat{a}_j^2 + \hat{a}_j^{\dagger 2} - \hat{a}_j \hat{a}_i - \hat{a}_j^\dagger \hat{a}_i^\dagger \right) \right). \tag{13d}$$

This Hamiltonian from Equation (13a) describes tightly trapped ions interacting via a Coulomb potential.

To simulate this system with particles interacting via a waveguide, we compared the terms of $H_{\text{coul}}$ with the Hamiltonian from Equation (5a). We found that the individual terms agree if we choose distances $d_{ij}$, wavenumbers $k$ and interaction strengths $\Omega_k$ such that:

$$\frac{\delta_0'}{4\pi\epsilon_0\hbar\omega'_T} \frac{q^2}{D_{ij}^2} = -\sum_k \frac{\epsilon_k}{\omega_T} \cos(kd_{ij}), \tag{14a}$$

$$\frac{\delta_0'^2}{2\pi\epsilon_0\hbar\omega'_T} \frac{q^2}{D_{ij}^3} = -\sum_k \frac{\Omega_k}{\omega_T} \sin(kd_{ij}). \tag{14b}$$

Note that the distances between the particles in our system, $d_{ij}$ are different from the distances in the simulated Coulomb system, $D_{ij}$. Additionally, the harmonic trapping frequencies $\omega_T$ and $\omega'_T$ and the respective trap widths $\delta_0$ and $\delta_0'$ can be chosen independently.

For given distances $d_{ij}$ and frequencies $k_l = k_0 + l\Delta_k$, Equation (14) becomes a set of linear equations for the interaction strengths $\Omega_l \equiv \Omega_{k_l}$. Choosing $D_{12} = D$ as a reference, we defined an interaction strength $\widetilde{\Omega}$ such that

$$\frac{q^2 \delta_0'^2}{2\pi\epsilon_0\hbar\omega'_T} \frac{1}{D^3} = \frac{\widetilde{\Omega}}{\omega_T}. \tag{15}$$

Equation (14) then simplifies to

$$\frac{D^3}{2D_{ij}^2} = -\delta_0' \sum_l \frac{\epsilon_l}{\widetilde{\Omega}} \cos((k_0 + l\Delta_k)d_{ij})$$

$$= -\frac{\delta_0'}{\delta_0} \sum_l \frac{\Omega_l}{\widetilde{\Omega}(k_0 + l\Delta_k)} \cos((k_0 + l\Delta_k)d_{ij}), \tag{16a}$$

$$\frac{D^3}{D_{ij}^3} = -\sum_l \frac{\Omega_l}{\widetilde{\Omega}} \sin((k_0 + l\Delta_k)d_{ij}). \tag{16b}$$

These equations form a linear system of equations for the interaction strength depending on the intensities of the incoming fields. Consequently, we have two equations for every different distance $D_{ij}$ and thus need the same number of fields. One way to solve this system is to assume that the particles are equally distributed with $d_{i,i+1} = 3\lambda_0/8$ and then choose the frequency spacing $\Delta_k$ such that the intensities $I_k$ are positive.

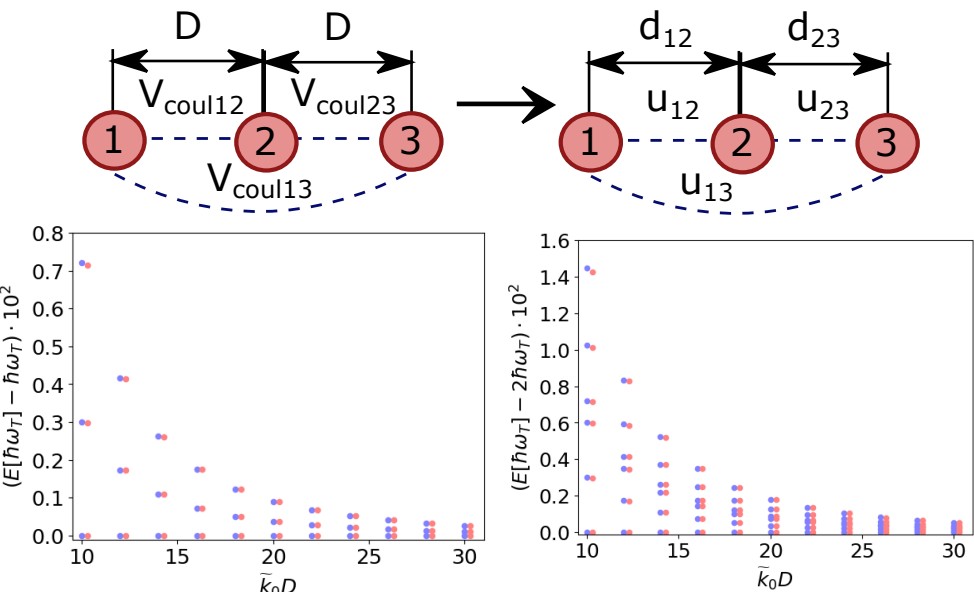

**Figure 2.** Schematic mapping of three ions along a line, interacting via the Coulomb force separated at a distance $D$ onto a system of particles along a nanofiber, with distances $d_{12}$ and $d_{23}$. This can be achieved by finding the right distances, frequencies and interactions strengths to solve Equations (16). The lower figures show the eigenenergies of these three harmonically trapped ions with Coulomb repulsion (blue) in comparison to particles trapped along the fiber with simulated interaction (red) as a function of the distance between the interacting particles. In the simulation, the particles are trapped along the fiber at fixed distances $d_{12} = d_{23} = 3/8\,\lambda_0$ and the pump laser parameters are adjusted to mimic Coulomb interaction at arbitrary distances between the ions. We used $\Delta_k = 0.7\,k_0$ and $\Omega_0/\widetilde{\Omega} = 0.34 + 1.07\,\widetilde{k}_0 D$, $\Omega_1/\widetilde{\Omega} = 1.16 + 1.34\,\widetilde{k}_0 D$, $\Omega_2/\widetilde{\Omega} = 1.68 + 1.58\,\widetilde{k}_0 D$, $\Omega_3/\widetilde{\Omega} = 0.74 + 1.4\,\widetilde{k}_0 D$, with $\widetilde{k}_0 = k_0\delta_0/\delta_0'$. For this figure, we chose $\widetilde{\Omega}/\omega_T$, such that every $\Omega_l/\omega_T \leq 0.004$ is restricted as required by Equation (11), that is, $\widetilde{\Omega}/\omega_T = \max_l(\Omega_l/\widetilde{\Omega})/(0.004\widetilde{k}_0 D)^3$. The figure on the left side shows the energies corresponding to the first oscillator state and the right figure shows the energies corresponding to the second oscillator state.

Although the fiber system is a 1D system, one could even map 2D or 3D systems on it. In this case, we have $N \cdot N_D$ oscillators, with $N$ being the number of ions and $N_D$ the number of dimensions. In general, this means we also need $N \cdot N_D$ particles in the simulation. Here, the first $N$ particles correspond to the interactions between the ions in the first dimension and the second $N$ particles correspond to the interactions in the second dimension and so on.

Figure 3 shows three ions arranged in an equilateral triangle. In this case, we have $N_D = 2$ dimensions and $N = 3$ ions and thus need six particles along the fiber. The Coulomb potential is then given by

$$V_{\text{coul}} = \sum_{i=1}^{N}\sum_{i\neq j} \frac{1}{8\pi\epsilon_0}\frac{q_i q_j}{\sqrt{D_{ij_x}^2 + D_{ij_y}^2}}, \tag{17}$$

with $D_{ij_x}$ and $D_{ij_y}$ being the distances between the ions in the $x$ and $y$ directions and $D_{ij}^2 = D_{ij_x}^2 + D_{ij_y}^2$. Here, we have to linearize in the $x$ as well as in the $y$ direction. We

assumed that the oscillators in the $x$ and $y$ directions have the same trapping frequency $\omega'_T$. To reduce the number of equations, we ignored the fast oscillating terms $\hat{H}_{\text{coul}_{\text{rwa}}}$ and found

$$\hat{H}_{\text{coul}} = \hat{\bar{H}}_{\text{coul}_{\text{osc}}} + \hat{H}_{\text{coul}_{\text{int}}}, \tag{18a}$$

$$\hat{\bar{H}}_{\text{coul}_{\text{osc}}} = \sum_{i=1}^{N} \hbar \left( \omega'_T + \frac{1}{8\pi\epsilon_0 \hbar} \sum_{j \neq i} \frac{4q^2 \delta_0'^2}{D_{ij}^5} \right) \left( \left( D_{ij_x}^2 - \frac{1}{2} D_{ij_y}^2 \right) \hat{a}_{i_x}^\dagger \hat{a}_{i_x} \right.$$

$$\left. + \left( D_{ij_y}^2 - \frac{1}{2} D_{ij_x}^2 \right) \hat{a}_{i_y}^\dagger \hat{a}_{i_y} \right), \tag{18b}$$

$$\hat{H}_{\text{coul}_{\text{int}}} = -\frac{1}{8\pi\epsilon_0} \sum_{i=1}^{N} \sum_{j \neq i} \frac{4q^2 \delta_0'^2}{D_{ij}^5} \left( \left( D_{ij_x}^2 - \frac{1}{2} D_{ij_y}^2 \right) \hat{a}_{i_x}^\dagger \hat{a}_{j_x} + \left( D_{ij_y}^2 - \frac{1}{2} D_{ij_x}^2 \right) \hat{a}_{i_y}^\dagger \hat{a}_{j_y} \right.$$

$$\left. - \frac{3}{2} D_{ij_x} D_{ij_y} \left( \hat{a}_{i_x}^\dagger \hat{a}_{j_y} + \hat{a}_{i_x} \hat{a}_{j_y}^\dagger - \hat{a}_{i_x} \hat{a}_{i_y}^\dagger - \hat{a}_{i_x}^\dagger \hat{a}_{i_y} \right) \right). \tag{18c}$$

Defining an interaction strength $\widetilde{\Omega}$, $D = D_{12}$ and $k_l = k_0 + l\Delta_k$ as in Equation (15), we have to solve the following 15 equations

$$\frac{D^3}{D_{ij}^5} \left( D_{ij_x}^2 - \frac{1}{2} D_{ij_y}^2 \right) = -\sum_l \frac{\Omega_l}{\widetilde{\Omega}} \sin((k_0 + l\Delta_k) d_{i_x j_x}), \tag{19a}$$

$$\frac{D^3}{D_{ij}^5} \left( D_{ij_y}^2 - \frac{1}{2} D_{ij_x}^2 \right) = -\sum_l \frac{\Omega_l}{\widetilde{\Omega}} \sin((k_0 + l\Delta_k) d_{i_y j_y}), \tag{19b}$$

$$\frac{3}{2} \frac{D^3}{D_{ij}^5} D_{ij_x} D_{ij_y} = \sum_l \frac{\Omega_l}{\widetilde{\Omega}} \sin((k_0 + l\Delta_k) d_{i_x j_y}), \tag{19c}$$

$$\sum_j \frac{3}{2} \frac{D^3}{D_{ij}^5} D_{ij_x} D_{ij_y} = -\sum_l \frac{\Omega_l}{\widetilde{\Omega}} \sin((k_0 + l\Delta_k) d_{i_x i_y}). \tag{19d}$$

A special feature of such a mapping is that the interactions between specific pairs of particles can be individually tuned or even turned off, as in Figure 3, for the particles at the bottom of the triangle. This allows one to implement any graph of interacting particles. Obviously, such systems could not be implemented with actual ions.

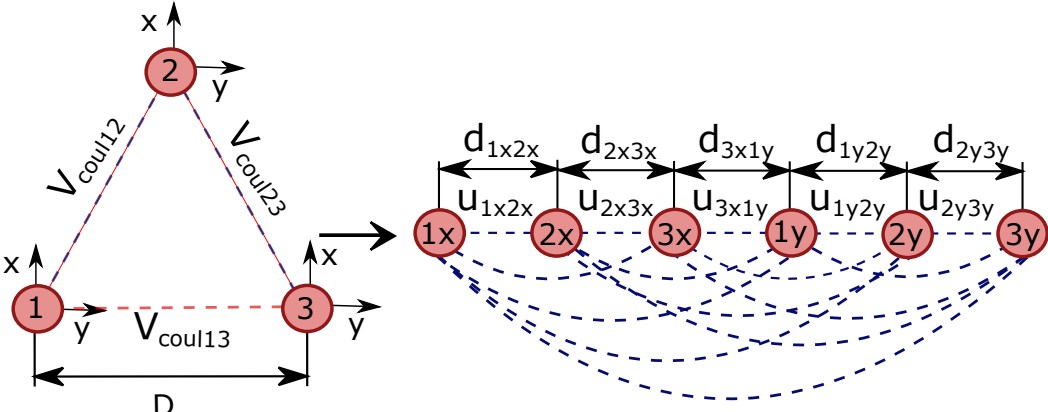

**Figure 3.** We simulated the Coulomb interaction of three ions arranged in an equilateral triangle by a 1D particle chain along a nanofiber. The first three particles correspond to the interaction in the $x$ direction, while the next three particles correspond to the interaction in the $y$ direction. In Figure 4, we also show an example where the interaction between the ion numbers 1 and 3 is turned off.

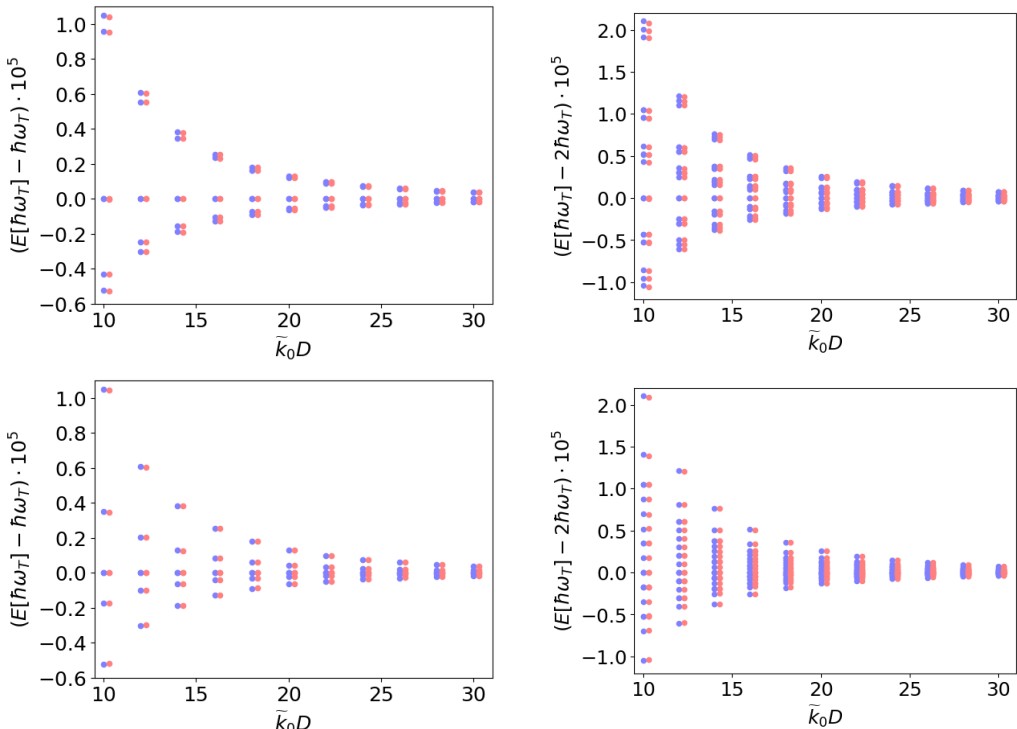

**Figure 4.** Eigenenergies of three harmonically trapped ions with Coulomb repulsion ordered like a triangle (blue) in comparison to particles trapped along a fiber with light-induced interaction (red), as shown in Figure 3, as a function of the distance between the interacting ions. This can be achieved by finding the right distances, frequencies and interaction strengths to solve Equation (19). In the simulation, the particles are trapped along the fiber at fixed distances $d_{12} = d_{23} = 1/3 \, \lambda_0$, $d_{34} = \lambda_0$ and $d_{45} = d_{56} = 1/4 \, \lambda_0$ and the pump laser parameters are adjusted to mimic Coulomb interactions at arbitrary distances between the ions with $\Delta_k = 0.33 \, k_0$. $\widetilde{k}_0$ and $\widetilde{\Omega}/\omega_T$ are defined as in Figure 2. The figures in the upper row show the eigenenergies when all particles are interacting, while in the lower figures the interaction between ion numbers 1 and 3 at the bottom of the triangle is suppressed. The figures on the left side show the energies corresponding to the first oscillator state and the right figures show the energies corresponding to the second oscillator state. Data values can be found in the Appendix A in Table A1.

To show the validity of this mapping, we calculated the eigenenergies of the waveguide system and compared them with the original Coulomb system (cf. Figure 2 for the ions along a line and Figure 4 for the ions ordered like a triangle). The energy levels of the oscillators split up due to the interaction between the particles. As long as the tight-binding condition for the ions was met, we found excellent agreement for all three systems. Note that the frequencies to solve Equations (16) and (19) are distributed over a large spectrum between $k_0$ and 3.8 $k_0$ and $k_0$ and 5.3 $k_0$, respectively. However, other methods to solve the equations might avoid this issue.

### 3.2. Bipartite Quantum Gates between Distant Particles

In the previous section, we showed how changing the distances between the traps or the intensity and frequency of the incoming light fields allows one to tailor the interaction between the particles. Here, we demonstrate how this can be used to design quantum gates.

Writing the Hamiltonian from Equation (5a) in an interaction picture, we find that the terms $\propto \hat{a}_i$, $\hat{a}_i^\dagger$ oscillate with $\tilde{\omega}_i$ and the terms $\propto \hat{a}_i^2$, $\hat{a}_i^{\dagger 2}$, $\hat{a}_j \hat{a}_i$, $\hat{a}_j^\dagger \hat{a}_i^\dagger$ oscillate with $2\tilde{\omega}_i$, while

the terms $\propto \hat{a}_j^\dagger \hat{a}_i$, $\hat{a}_j^\dagger \hat{a}_j$ do not oscillate. The rapidly oscillating terms average to zero and thus the Hamiltonian of Equation (5a) simplifies to

$$\hat{H}_{int} = \sum_{j=1}^{N} \sum_{i=1}^{N} \sum_{k} \hbar \Omega_k \sin\left(k|x_{j,0} - x_{i,0}|\right) a_j^\dagger a_i = \sum_{i,j=1}^{N} \hbar g_{ij} a_j^\dagger a_i, \tag{20}$$

with $\Omega_k = \sigma_{sc} I_k \delta_0^2 / (\hbar k c)$ and $g_{ij} := \sum_k \Omega_k \sin(k d_{ij})$. Now it is obvious that the interaction between any particle pair $i$ and $j$ can be turned off by finding frequencies, positions and intensities such that the coupling $g_{ij}$ vanishes. This is very important for gates as they should only act on special particles and not on all of them.

Figure 5 shows an example of three particles, where only two particles interact. Choosing the distances and frequencies with $k_l = k_0 + l \Delta_k$ and $\Delta_k < k_0$, we have to solve $N(N-1)/2$ equations, with $N$ being the number of the particles in the system, and thus need $N(N-1)/2$ frequencies and intensities. Note that the distances have to be chosen such that they are different for interacting and non-interacting pairs.

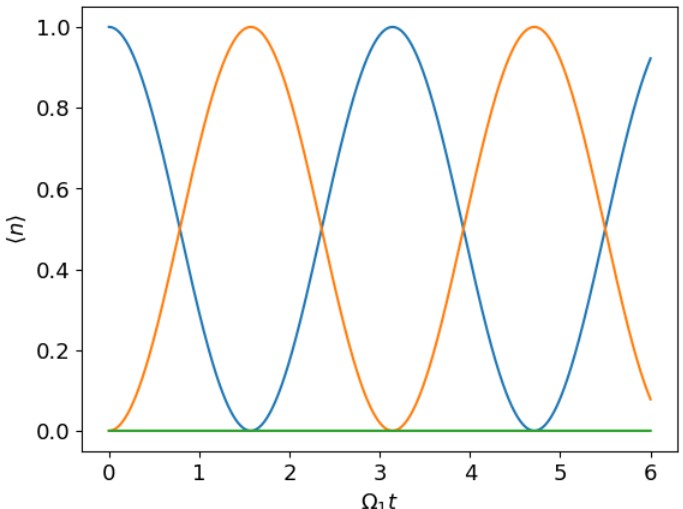

**Figure 5.** Time evolution of the excited motional state occupation for three coupled particles. The blue line corresponds to the first particle, the orange line to the second particle and the green line to the third particle. We start from a state $|100\rangle$, where only the first particle is excited, and set the distances to $d_{12} = 3/4\,\lambda_1$, $d_{23} = 7/8\,\lambda_1$, $k_2 = 4/3 k_1$ and $\Omega_2 = 0.82\,\Omega_1$. Choosing these parameters, the interaction between the third and the other two particles can be turned off, while the first two particles still interact with each other. Deactivating the interaction between specific particle pairs is necessary to implement bipartite quantum gates.

3.2.1. Using the Two Lowest Oscillator States on a Qubit Basis

The simplest way to map the harmonic oscillator to a qubit system is to consider only the ground $|0\rangle$ and first excited state $|1\rangle$. As the Hamiltonian from Equation (20) is $\propto a_j^\dagger a_i$, the resulting dynamics do not leave the subspace of these two states. For the basis $(|0\rangle|0\rangle, |0\rangle|1\rangle, |1\rangle|0\rangle$ and $|1\rangle|1\rangle)$, the states evolve with the time-evolution operator $\hat{U}(t) = \exp(-i\hat{H}t/\hbar)$,

$$\hat{U}(t) = \begin{pmatrix} 1 & 0 & 0 & 0 \\ 0 & \cos(2gt) & -i\sin(2gt) & 0 \\ 0 & -i\sin(2gt) & \cos(2gt) & 0 \\ 0 & 0 & 0 & 1 \end{pmatrix}, \tag{21}$$

which is equivalent to a mapping

$$|0\rangle|0\rangle \to |0\rangle|0\rangle \tag{22a}$$
$$|0\rangle|1\rangle \to \cos(2gt)|0\rangle|1\rangle - i\sin(2gt)|1\rangle|0\rangle \tag{22b}$$
$$|1\rangle|0\rangle \to -i\sin(2gt)|0\rangle|1\rangle + \cos(2gt)|1\rangle|0\rangle \tag{22c}$$
$$|1\rangle|1\rangle \to |1\rangle|1\rangle. \tag{22d}$$

In this case, the states $|0\rangle|0\rangle$ and $|1\rangle|1\rangle$ are not affected by the interaction, but we see oscillations between $|0\rangle|1\rangle$ and $|1\rangle|0\rangle$.

After an interaction time such that $gt = \pi/4 + 2\pi n$, $n \in \mathbb{Z}$, $U(t)$ changes to

$$\hat{U}_{\text{SWAP}} = \begin{pmatrix} 1 & 0 & 0 & 0 \\ 0 & 0 & -i & 0 \\ 0 & -i & 0 & 0 \\ 0 & 0 & 0 & 1 \end{pmatrix}. \tag{23}$$

This corresponds to an i-SWAP gate, which swaps the states of the two particles and introduces a phase if the two particles are in different states.

Similarly, the square root of an i-SWAP gate (SQiSW) can be implemented by choosing $g_{ij}t = \pi/8 + \pi n$, $n \in \mathbb{Z}$. Then, $U(t)$ changes to

$$\hat{U}_{\text{SQiSW}} = \begin{pmatrix} 1 & 0 & 0 & 0 \\ 0 & \frac{1}{\sqrt{2}} & -\frac{i}{\sqrt{2}} & 0 \\ 0 & -\frac{i}{\sqrt{2}} & \frac{1}{\sqrt{2}} & 0 \\ 0 & 0 & 0 & 1 \end{pmatrix}. \tag{24}$$

$\hat{U}_{\text{SQiSW}}$ is a universal entangling gate and any quantum computation can be implemented using only single qubit rotations and the SQiSW gate [44]. However, note that single qubit rotations cannot be implemented in the formalism described here as every interaction changes the state of (at least) two particles. One would thus need a separate mechanism to rotate the state of each particle individually.

### 3.2.2. Coherent States as a Computational Basis

As our individual quantum systems are oscillators, we can go beyond the two-state approximations and also use higher excited motional states as a computational basis. One particularly useful approach, which has been put forward and intensively studied for photons, is the use of coherent states as qubits. Typically, the computational basis is a pair of coherent states $|-\alpha\rangle$, $|+\alpha\rangle$ [45–48],

$$|\alpha\rangle = \sum_{i=0}^{\infty} e^{-\frac{|\alpha|^2}{2}} \frac{\alpha^i}{\sqrt{i!}} |i\rangle, \tag{25}$$

with a complex amplitude $\alpha$. Although the two states are not perfectly orthogonal, the overlap between $|+\alpha\rangle$ and $|-\alpha\rangle$ is negligibly small for a sufficiently large $|\alpha|$. For example,

$$|\langle +\alpha| - \alpha\rangle|^2 = e^{-4|\alpha|^2} \approx 0.018, \tag{26}$$

for amplitudes as small as $\alpha = 1$. On this basis, quantum calculations can be performed that are relatively loss and fault tolerant [47], and it turns out that all relevant two-qubit interactions can be based on the so-called beamsplitter coupling between two sites [48]. The interaction is then simply given by $U(t) = \exp\left(i\theta/2\left(\hat{a}_1^\dagger \hat{a}_2 + \hat{a}_1 \hat{a}_2^\dagger\right)\right)$, with $\theta$ the polarization angle between the two interacting beams. It turns out that this is just the dominant term of light scattering interaction (20), which can be well controlled in terms of strength, time and space.

The subset of coherent states $\{|\alpha\rangle|\alpha'\rangle\}$, with $|\alpha| = |\alpha'|$ evolves as

$$e^{i\hat{H}_{int}t/\hbar}|\alpha\rangle|\alpha'\rangle = |\alpha\cos(gt) + i\alpha'\sin(gt)\rangle|\alpha'\cos(gt) + i\alpha\sin(gt)\rangle. \quad (27)$$

such that

$$|-\alpha\rangle|-\alpha\rangle \rightarrow |-e^{igt}\alpha\rangle|-e^{igt}\alpha\rangle \quad (28a)$$

$$|-\alpha\rangle|+\alpha\rangle \rightarrow |-e^{-igt}\alpha\rangle|e^{-igt}\alpha\rangle \quad (28b)$$

$$|+\alpha\rangle|-\alpha\rangle \rightarrow |e^{-igt}\alpha\rangle|-e^{-igt}\alpha\rangle \quad (28c)$$

$$|+\alpha\rangle|+\alpha\rangle \rightarrow |e^{igt}\alpha\rangle|e^{igt}\alpha\rangle \quad (28d)$$

A more detailed calculation of this evolution can be found in Appendix B. As discussed in [47], this evolution corresponds to a beamsplitter interaction for photonic states. There it is also discussed that one can use this and a single-qubit rotation to implement a CNOT gate.

In contrast to what we found in Equation (22), we see here that the state $|-\alpha\rangle|-\alpha\rangle$ can be flipped to $|+\alpha\rangle|+\alpha\rangle$ and vice versa. Note also that the coherent qubits evolve outside the subspace $\{|+\alpha\rangle, |-\alpha\rangle\}$ for $gt \neq n\pi$.

### 3.3. Entanglement Propagation via Controlled Long-Range Interaction

The discussion above focused on entangling any two particles in a larger system using quantum gates realized by two-particle gates. Here, we shall briefly investigate how a larger number of particles can be entangled.

If we only have a single pump field of frequency $k_0$ and put all particles at equal distance $n\pi k_0$, with an arbitrary integer $n$, then no particle will interact with any other particle as $\sin(k_0 d_{ij}) = 0$. If we now displace one particle by $\zeta \neq n\pi k_0$, this particle starts to interact with all other particles, but there are still no direct interactions between the remaining particles.

However, as shown in Figure 6, this is sufficient to create an effective all-to-all interaction. In this figure, there are three particles where the first and the second particles do not interact directly, but both interact with the third particle.

This is demonstrated using the mutual von-Neumann entropy,

$$S_i = -\text{tr}(\rho_i \ln \rho_i), \quad (29)$$

with $\rho_i$ being the reduced density matrix of the subsystem $i$.

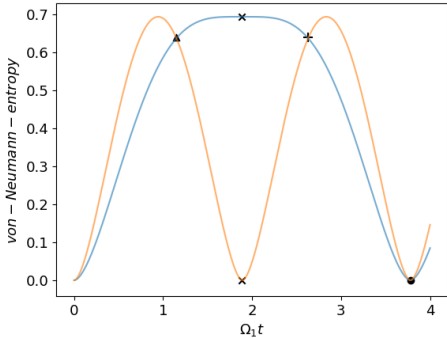

**Figure 6.** Entanglement propagation for three particles coupled via a single illumination beam as a function of time with $d_{12} = \lambda_0/2$ and $d_{23} = (1/2 + 0.1)\lambda_0$ for the initial state $|001\rangle$. The two curves show the entanglement entropy of the subsystem containing particles 2 and 3 (blue) and the subsystem containing particles 1 and 2 (yellow). So, the blue line describes the entanglement between the subsystem containing particle 1 and the subsystem containing particles 2 and 3, and the yellow line between the subsystem containing particle 3 and the subsystem containing particles 1 and 2. (▲) corresponds to the state $1/\sqrt{3}(|001\rangle - i|010\rangle + i|100\rangle)$, (×) to the state $1/\sqrt{2}(|01\rangle - |10\rangle)|0\rangle$, (+) to the state $\frac{1}{\sqrt{3}}(-|001\rangle + i|010\rangle - i|100\rangle)$, and (•) to the state $-|001\rangle$.

In the left plot of Figure 6, we start with a pure state, $|001\rangle$. However, after a time such that $\cos\left(2\sqrt{2}\Omega_1 \sin(k\zeta)t\right) = 1/\sqrt{3}$, indicated by the triangle, all three particles are entangled. Later, at the time indicated by the ($\times$), particles one and two are maximally entangled with each other but disentangled from the third particle.

### 3.4. State Read Out via the Outgoing Fiber Fields

In the previous chapters we discussed how the motional states of the particles can be manipulated, but how would such a manipulation be measured?

The fields leaving the fiber at the left and right edges contain information about the states of the particles in the system and by measuring the outgoing intensities one can determine the states of the particles.

Following the beamsplitter matrix formalism introduced in [19] we find for the amplitudes of the outgoing fields to the left $E_-(x_1)$ and to the right $E_+(x_N)$ of a system with $N$ particles,

$$E_-(x_1) = \sum_k \sum_{i=1}^{N} \sqrt{\frac{I_k}{c\epsilon_0}} e^{ik(x_i - x_1)}, \tag{30a}$$

$$E_+(x_N) = \sum_k \sum_{i=1}^{N} \sqrt{\frac{I_k}{c\epsilon_0}} e^{ik(x_N - x_i)}. \tag{30b}$$

As the particles are well trapped, we can linearize these amplitudes as we did for the Hamiltonian and find

$$\hat{E}_-(x_1) = \sum_k \sum_{i=1}^{N} \sqrt{\frac{I_k}{c\epsilon_0}} \left( e^{ik(x_i - x_1)} + ik\delta_0 e^{ik(x_i - x_1)} \left( \hat{a}_i + \hat{a}_i^\dagger - \hat{a}_1 - \hat{a}_1^\dagger \right) \right.$$
$$\left. -\frac{1}{2}k^2\delta_0^2 e^{ik(x_i - x_1)} \left( \hat{a}_i + \hat{a}_i^\dagger - \hat{a}_1 - \hat{a}_1^\dagger \right)^2 \right), \tag{31a}$$

$$\hat{E}_+(x_N) = \sum_k \sum_{i=1}^{N} \sqrt{\frac{I_k}{c\epsilon_0}} \left( e^{ik(x_N - x_i)} + ik\delta_0 e^{ik(x_N - x_i)} \left( \hat{a}_N + \hat{a}_N^\dagger - \hat{a}_i - \hat{a}_i^\dagger \right) \right.$$
$$\left. -\frac{1}{2}k^2\delta_0^2 e^{ik(x_N - x_i)} \left( \hat{a}_N + \hat{a}_N^\dagger - \hat{a}_i - \hat{a}_i^\dagger \right)^2 \right). \tag{31b}$$

This way, we can calculate the expectation values for amplitudes and intensities for any given particle state.

Figure 7 shows an example for the outgoing intensity expectation values for three particles. It confirms that the outgoing intensity depends on the states of the particles. Here, the states $|100\rangle$ and $|011\rangle$ cannot be distinguished in the left outgoing intensity, $I_-$, but they can be distinguished in $I_+$.

In Figure 8, we plotted the outgoing intensity for the initial conditions as in Figure 6. From the outgoing intensity, we can learn which particles are entangled and which are not.

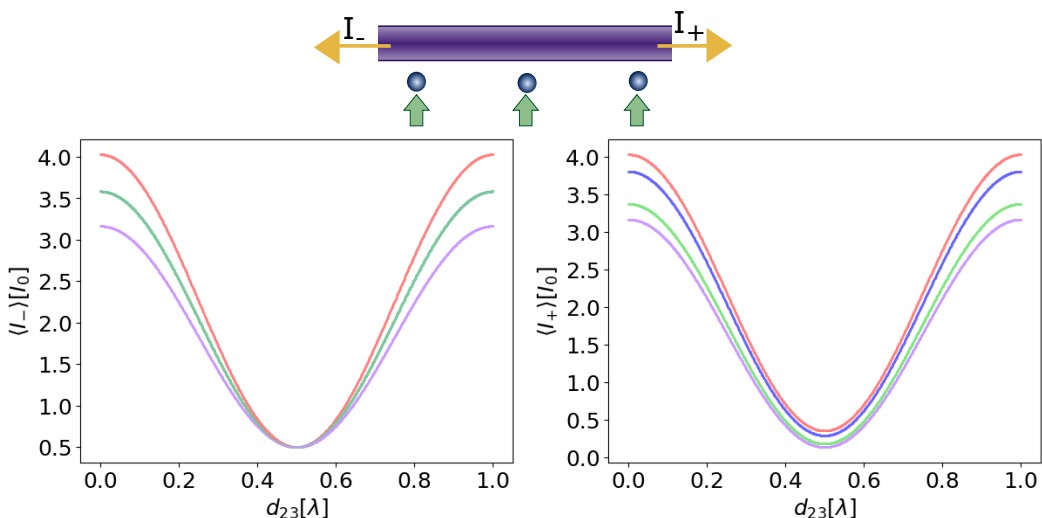

**Figure 7.** State-dependent light intensities emitted from the fiber to the left $I_-$ and to the right $I_+$ for a system with three particles. Here, the distance between the first two particles stays constant with $d_{12} = \lambda_0$, while we vary the position of the third particle. Red lines correspond to the ground state $|000\rangle$, blue lines to the single excited state $|100\rangle$, green lines to a doubly excited state $|011\rangle$ and purple lines corresponds to the state $|111\rangle$. Note that in the left figure the green and blue lines overlap.

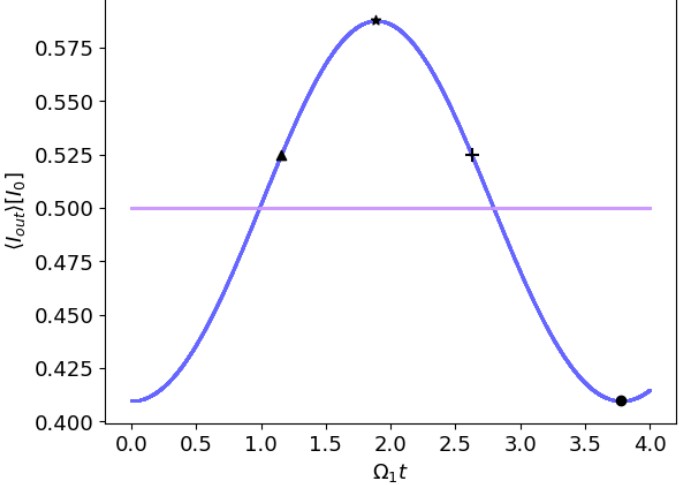

**Figure 8.** Average output power $I_-$ emitted on the left side of the fiber and $I_+$ on the right side for a system of two particles as a function of time. The initial condition is the same as in Figure 6. The blue line corresponds to the outgoing intensity to the left side $\langle I_- \rangle$ and the purple line to the (constant) outgoing intensity to right $\langle I_+ \rangle$. (▲) corresponds to the state $\frac{1}{\sqrt{3}}(|001\rangle - i|010\rangle + i|100\rangle)$, (★) to the state $\frac{1}{\sqrt{2}}(|01\rangle - |10\rangle|0\rangle$, (+) to the state $\frac{1}{\sqrt{3}}(-|001\rangle + i|010\rangle - i|100\rangle)$ and (·) to $-|001\rangle$. The light leaving the system thus contains information about the entangled motional states of the particles.

## 4. Discussion

Our theoretical studies suggest that transversely illuminated particles trapped next to a nanofiber offer a versatile platform for studying and testing a wide range of physical phenomena. In Section 3.1, we show that one is able to set up simulations for a wide range of two-particle interactions. Alternatively, possible implementations of bipartite quantum gates are demonstrated in Section 3.2, where one can switch from a two-state to an oscillator basis. Certainly, there are still some technical challenges to be met such as, for example, ensuring sufficient radial confinement and cooling.

To implement complex shaped interactions, one can make use of a range of illumination frequencies. A central challenge here of course is, as in the example discussed in

Section 3.1, that one needs a high number of frequencies over a large bandwidth to precisely control the interactions between the particles. In an experiment, one is generally restricted to frequencies within a certain bandwidth to ensure sufficient interaction between the atoms and the light field, which limits the system versatility.

Another practical issue to resolve is the finite back scattering of the fiber field by the particles, which we assumed to be negligible to arrive at Equation (1). Here, residual reflection inside the fiber opens additional coupling and interaction channels between the particles, which have to be accounted for, especially for larger particle numbers. The concrete implications are hard to predict in general and will have to be studied in detail for any specific experimental implementation.

As has been observed in some of the pioneering experiments, vibrations of the nanofiber can heat the particles and affect both the trapping of particles and their interactions. Luckily, to a large extent, this problem can be overcome by choosing the fiber radius to be as large as possible and by optimizing the taper at both ends of the nanofiber. In this case, one can choose special positions of the particles along the waveguide to mimimize vibration effects [49].

Theoretically, a more precise model should also include back scattering inside the fiber and a chirality-related scattering control [8,50–53]. This, on the one hand, could be very challenging to calculate, but, on the other hand, could be a very promising extension towards even more control possibilities of this system.

## 5. Conclusions

This work shows how mechanical interactions of particles trapped in the vicinity of an optical nanofiber can be controlled in a versatile form by choosing the properties of incoming transverse pump light. Using spatial and spectral light shaping of the illumination lasers, the interactions between the particles can be tailored to simulate a wide class of interaction potentials between the particles.

We studied the low temperature limit using a quantum mechanical description of the particle motion along the fiber direction at the trap sites and coupled the particles via a non-local interaction through collective coherent light scattering into the fiber. We demonstrate that this system can be used to simulate, for example, Coulomb interactions between harmonically trapped particles with high precision. The idea can be extended in a straightforward manner beyond linear equidistant chains to effectively mimic a very general class of geometries including 2D-configurations. Using time-dependent laser illumination, one can even turn on and off specific interactions between arbitrary particle pairs simultaneously. By monitoring the spectrum and intensity of the light scattered out of the fiber ends, ample information on the particle motion can be extracted in a minimally invasive way.

As another natural application, the system offers varied possibilities to design two-qubit gates, using not only oscillator eigenstates but also coherent states as a computational basis. The virtually infinite range of the fiber mediated interaction should allow us to implement larger systems of many qubits, without the requirement of closely spaced trapping sites, allowing independent pairwise addressing control of quantum gates. As generic examples, we studied the preparation of multi-partite entangled states by placing the particles at specific positions with respect to the illumination lasers. Again, monitoring the outgoing intensity at the fiber ends allows one to continuously determine key properties of the collective motional states of the particles with minimal perturbation of the entanglement properties.

So far, most experimental and theoretical works focused on the manipulation of and the coupling between internal atomic degrees of freedom. However, with the recent implementation of transverse pumping near a nanofiber [6], we believe the path to manipulating motional degrees of freedom is promising and open. Note that the general form of the interaction bears great similarity with the phonon-based motion interaction in ion traps, which proved to be one of the most successful quantum bus systems. Here, due to the

virtually infinite range of the fiber-based forces, one can choose a much larger particle spacing and thus separate individual and collective controls. In addition, the Coulomb interaction cannot simply be turned off as pump lasers can.

**Author Contributions:** Conceptualization, D.H.; methodology, D.H., M.S. and H.R.; formal analysis, D.H.; investigation, D.H.; writing—original draft preparation, D.H. and M.S.; writing—review and editing, D.H., M.S. and H.R.; visualization, D.H.; supervision, M.S. and H.R.; project administration, H.R. All authors have read and agreed to the published version of the manuscript.

**Funding:** D.H. was funded by a DOC Fellowship of the Austrian Academy of Sciences ÖAW and H.R. acknowledges support from the FET Network Cryst3 funded by the European Union (EU) via Horizon 2020.

**Conflicts of Interest:** The authors declare no conflict of interest.

## Appendix A. Data Values for Figure 4

Here, we list the distances, frequencies and interaction strengths used to create Figure 4 for different shapes of interacting ions.

**Table A1.** Data values for Figure 4.

|  | Triangle | Triangle with Suppressed Interactions |
| --- | --- | --- |
| $\Omega_0/\widetilde{\Omega}$ | 251.5 | 251.4 |
| $\Omega_1/\widetilde{\Omega}$ | 643 | 642.6 |
| $\Omega_2/\widetilde{\Omega}$ | 580.5 | 580.1 |
| $\Omega_3/\widetilde{\Omega}$ | 72 | 72 |
| $\Omega_4/\widetilde{\Omega}$ | 0 | 0 |
| $\Omega_5/\widetilde{\Omega}$ | 666.2 | 665.8 |
| $\Omega_6/\widetilde{\Omega}$ | 1149.7 | 1149.1 |
| $\Omega_7/\widetilde{\Omega}$ | 754.3 | 754.3 |
| $\Omega_8/\widetilde{\Omega}$ | 104.7 | 104.8 |
| $\Omega_9/\widetilde{\Omega}$ | 115.5 | 115.3 |
| $\Omega_{10}/\widetilde{\Omega}$ | 591.3 | 590.8 |
| $\Omega_{11}/\widetilde{\Omega}$ | 724.8 | 724.5 |
| $\Omega_{12}/\widetilde{\Omega}$ | 392.4 | 392.4 |
| $\Omega_{13}/\widetilde{\Omega}$ | 81.2 | 81.3 |

## Appendix B. Time Evolution of the Coherent States

In this section, we go into more detail on the time evolution of coherent states $|\alpha\rangle|\alpha'\rangle$

$$\hat{U}(t)|\alpha\rangle|\alpha'\rangle = e^{-igt\left(\hat{a}_1\hat{a}_2^\dagger + \hat{a}_1^\dagger\hat{a}_2\right)}|\alpha\rangle|\alpha'\rangle, \tag{A1}$$

with $\alpha$, $\alpha'$ taking values of $\pm\alpha$.

Using the definition of a coherent state, and the facts that $\hat{U}(t)|00\rangle = |00\rangle$ and that $\hat{U}(t)$ is unitary, we can rewrite this equation

$$\hat{U}(t)|\alpha\rangle|\alpha'\rangle = e^{-\frac{|\alpha|^2+|\alpha'|^2}{2}} \sum_{m,n=0}^{\infty} \frac{\alpha^m \alpha'^n}{\sqrt{m!n!}} \left(\hat{U}(t)\hat{a}_1^\dagger\hat{U}^\dagger(t)\right)^m \left(\hat{U}(t)\hat{a}_2^\dagger\hat{U}^\dagger(t)\right)^n |00\rangle. \tag{A2}$$

To evaluate $\hat{B}_i(t) := \hat{U}(t)\hat{a}_i^\dagger\hat{U}^\dagger(t)$, we use

$$\frac{d}{dt}\hat{B}_i(t) = ig\hat{U}(t)\left[\hat{a}_1^\dagger\hat{a}_2 + \hat{a}_1\hat{a}_2^\dagger, \hat{a}_i^\dagger\right]\hat{U}^\dagger(t) = ig\hat{A}_j, \tag{A3}$$

for $j \neq i$. The solution of this system of differential equations is

$$\hat{A}_i = \hat{a}_i^\dagger \cos(gt) + i\hat{a}_j^\dagger \sin(gt).\tag{A4}$$

This way, the sum in Equation (A2) can be rewritten as

$$\sum_{m,n=0}^{\infty} \frac{\alpha^m \alpha'^n}{\sqrt{m!n!}} \left(\hat{a}_1^\dagger \cos(gt) + i\hat{a}_2^\dagger \sin(gt)\right)^m \left(i\hat{a}_1^\dagger \sin(gt) + \hat{a}_2^\dagger \cos(gt)\right)^n |00\rangle$$

$$= e^{\alpha\left(\hat{a}_1^\dagger \cos(gt) + i\hat{a}_2^\dagger \sin(gt)\right)} e^{\alpha'\left(i\hat{a}_1^\dagger \sin(gt) + \hat{a}_2^\dagger \cos(gt)\right)} |00\rangle$$

$$= e^{\hat{a}_1^\dagger\left(\alpha \cos(gt) + i\alpha' \sin(gt)\right)} e^{\hat{a}_2^\dagger\left(\alpha' i \sin(gt) + \alpha \cos(gt)\right)} |00\rangle\tag{A5}$$

As $e^{-\alpha^\star\left(i\hat{a}\sin(gt) + \hat{a}\cos(gt)\right)}|0\rangle = |0\rangle$ and the displacement operator is defined as $\hat{D}(\alpha) = e^{-|\alpha|^2/2} e^{\alpha\hat{a}^\dagger} e^{-\alpha^\star\hat{a}}$; with $D(\alpha)|0\rangle = |\alpha\rangle$, we find the desired result

$$\hat{U}(t)|\alpha\rangle|\alpha'\rangle = \hat{D}_1\left(\alpha \cos(gt) + i\alpha' \sin(gt)\right)\hat{D}_2\left(\alpha' \cos(gt) + i\alpha \sin(gt)\right)|0\rangle|0\rangle$$

$$= |\alpha \cos(gt) + i\alpha' \sin(gt)\rangle|\alpha' \cos(gt) + i\alpha \sin(gt)\rangle.\tag{A6}$$

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
