# Peer review of "A Versatile Quantum Simulator for Coupled Oscillators Using a 1D Chain of Atoms Trapped near an Optical Nanofiber"

_photonics, doi:10.3390/photonics8060228_

Round 1

Reviewer 1 Report

In the work 'A VERSATILE QUANTUM SIMULATOR FOR COUPLED OSCILLATORS USING A 1D CHAIN OF ATOMS TRAPPED NEAR AN OPTICAL NANOFIBER', D. Holzmann and co-authors analyze the applications of a linear chain of atoms trapped by a strong harmonic potential close to an optical nanofiber.

The paper presents in a detailed form the technical calculations from which they extract the conditions to simulate Coulomb interactions from a set o linearly coupled oscillator equations. The mapping can be done exactly comparing the expressions of the Hamiltonians for the physical (eqs. 5) and model systems (eqs.13). The authors then proceed to discuss an application to the creation of multi-qubit quantum gates with tailored interactions and the state detection via intensity measurement of the outgoing light field from the optical fiber.

The presentation of the results is clear. The original findings build up from previous work from the same authors (refs. [18,20]) and they describe an experimentally feasible scheme. As a minor comment I would ask the authors to comment on the scalability of this scheme for what concerns the realization of Coulomb interactions for a larger system and what would the limitations be.

In summary, I recommend this work for publication. I would suggest some minimal changes to improve the presentation and a clarification to the question above.

Author Response

Thank you for the positive assessment and your comments. In response to your request for clarifications on the scalability of the system we now review several challenges to a possible experimental realization in the updated "Discussion" section. The most pressing issues are probably the need for many frequencies in the transverse laser beams and residual back-scattering of light inside the fiber which we neglect in our model.

Reviewer 2 Report

The authors have investigated the low-temperature limit of the particle motion along with the fiber.

They stated that they can extract in minimally invasive way information on the particle motion.

The subject is interesting however before giving my final decision, some critical remarks should be answered and taken into consideration for the revised version:

  • The definition of the Figures should be increased.
  • The paper needs to mention physical realizations from previous studies on similar effects.
  • Applications are only mentioned briefly. The applications should be discussed in the text and in the conclusion in detail.

Author Response

Thank you for the positive assessment and your comments.
As requested we modified the captions of Figures 2, 4, 5, 6 and 8 and hope that they are now easier to understand.
Following the request to mention more physical realizations we now found and cite a recent experiment by Meng el al. [1]. They recently managed to scatter light from a transverse pump into a fiber via atoms trapped next to the nanofiber. We consider this an important step towards an experimental implementation of our proposal.
Regarding the request to mention more applications we would like to highlight that our model is very versatile: In principle one could simulate any two-particle interaction (not just the Coulomb-force) and the quantum gates described in section 3.2 could be part of a universal quantum computer. This versatility is why we chose not to focus on one particular application and we hope our comments in the "Conclusion" section clarify the wide range of applications. 
[1] Y. Meng C. Liedl, S. Pucher, A. Rauschenbeutel, P. Schneeweiss: Imaging and localizing381individual atoms interfaced with a nanophotonic waveguide. Phys. Rev. Lett. 125, 053603 (2020).

Reviewer 3 Report

The paper presents the novel approach to simulate quantum many-body systems with arbitrary two-body interaction potential. The proposed implementation is based on the interaction of individual trapped atoms with the guided mode of an optical nanofiber. As an impressive example, the authors show that the appropriate choice of the positions of the atoms and intensities of the transverse pumps can simulate Coulomb interaction between charged particles. In addition, the coupling of particles via the nanofiber allows for the realization of quantum logic gates.

I find the presented results very interesting and scientifically sound. The theoretical analysis is performed with sufficient details, the values of the parameters for possible experimental implementation are estimated.

l have no doubts that the paper can be accepted in the present form. There are maybe a few items that the authors could explain in somewhat more detail:

  1. It is not completely clear from the presented discussion that the fiber mode degree of freedom can be adiabatically eliminated. It seems natural taking into account the propagating character of the fiber mode and small system size, but it might be useful if the authors add a few words about that. 
  2. Another question that seems to be connected to the previous item is: Why can the self-action of atoms be excluded? The atom scatters the pump field, which contributes to the fiber-mode intensity that in turn produces the force. Is it also due to the wave propagation in the fiber?
  3. I have some concerns about the technical difficulties of the implementation. In particular, the length of the nanofiber for a large number of atoms should be quite large, since one should provide enough space between the atoms to individually trap and pump them. Thus one can expect the existence of low-energy mechanical vibrating modes of the nanofiber. Such modes could be optomecanically excited due to pump fields. The resulting motion of the fiber can probably affect the forces acting on the atoms. It would be interesting to know the opinion of the authors if such an effect can be an obstacle for the experimental realization of the proposal.
  4. If I understand correctly there is a small notation inconsistency.  I_k in Eq. 1 seems to mean the intensity of the pump,  since this equation also contains the scattering cross section.  In Eq. 6 the same notation I_k is used for the scattered light intensity.

Author Response

Thank you for the positive assessment and your comments.
Remarks 1 & 2 on the fiber modes are indeed connected so we shall attempt to answer them together: In our model the particles only interact with the fiber mode in a way that they scatter light from the transverse pump into the fiber. The light scattered by particles at different positions can then interfere which, in the end, leads to an effective interaction between the particles. This model is comparable to light-induced dipole-dipole interaction in a rarefied gas where multiple reflections of the light between the particles can be neglected.
A classical calculation in the context of a previous work [1] showed this restricted model works well for dozens of particles. Beyond that the interaction between the particles and the fiber mode has to be fully included.
Following the referee's remarks we clarified this point of the model and its implications in the text above Equation (1). We hope that this answers the referee's questions. 
In his/her third question the referee mentions technical difficulties due to vibrations. In response we now include a reference to a recent work by D. Hümmer et al. [2]. They show that, as expected, vibrating modes can lead to heating of the particles. But the authors also suggest that using a larger fiber radius can reduce this effect.
A second approach is the optimization of the fiber taper in a way that these vibration modes are reflected. Then the heating rate depends on the position of the particles along the fiber and can be reduced by choosing special positions.
Regarding the fourth comment we would like to clarify that the intensity I_k is the intensity inside the fiber both in Eq. (6) and Eq. (1). The scattering cross section \sigma_{sc} enters as an effective area of the atom. In combination with the intensity inside the fiber, I_k, and the speed of light this gives a force on the particle. 
[1] D. Holzmann et al. New J. Phys. 20 103009 (2018)
[2] D. Hümmer, P. Schneeweiss, A. Rauschenbeutel, O. Romero-Isart: Heating in nano photonic traps for cold atoms. Phys. Rev. X 9, 041034 (2019)

Reviewer 4 Report

This manuscript provides sufficient theoretical insights and in-depth analysis of the results. 

Author Response

Thank you for your positive assessment of our work.

Round 2

Reviewer 2 Report

The revised version is ok. I accept it.